# Nutrition and Diet in the Prevention and Management of Prostate Cancer in Mexico: A Narrative Review

**DOI:** 10.3390/nu17132151

**Published:** 2025-06-27

**Authors:** Sarai Citlallic Rodríguez Reyes, Cecilia Rico Fuentes, Ana Laura Pereira Suárez, Erick Sierra Díaz, José Miguel Moreno Ortíz, Adrián Ramírez de Arellano

**Affiliations:** 1Instituto de Nutrigenética y Nutrigenómica Traslacional, Centro Universitario de Ciencias de la Salud, Universidad de Guadalajara, Guadalajara 44340, Mexico; citlalic.rodriguez@academicos.udg.mx; 2Laboratorio de Investigación en Cáncer e Infecciones, Departamento de Microbiología y Patología, Centro Universitario de Ciencias de la Salud, Universidad de Guadalajara, Guadalajara 44340, Mexico; cecilia.rico9313@alumnos.udg.mx (C.R.F.); ana.pereira@academicos.udg.mx (A.L.P.S.); 3Departamento de Salud Pública, Centro Universitario de Ciencias de La Salud, Universidad de Guadalajara, División de Epidemiología, Unidad Medica de Alta Especialidad, Hospital de Especialidades, Centro Médico Nacional de Occidente, Guadalajara 44340, Mexico; erick.sierra@imss.gob.mx; 4Instituto de Genética Humana “Dr. Enrique Corona Rivera”, Departamento de Biología Molecular y Genómica, Centro Universitario de Ciencias de la Salud, Universidad de Guadalajara, Guadalajara 44340, Mexico; miguel.moreno@academicos.udg.mx

**Keywords:** prostate cancer, nutrition, prevention, Mexican population

## Abstract

Prostate cancer is the most frequently diagnosed cancer among men and represents a significant public health challenge, particularly in Mexico, where it is the second leading cause of cancer-related mortality. Early prevention strategies are urgently needed due to the disease’s aggressive progression and its impact on long-term survival. Nutritional interventions have garnered increasing attention, especially in light of risk factors such as aging, obesity, and adipose tissue dysfunction, which contribute to elevated prostate cancer risk. As incidence continues to rise among men over 50, promoting diet-based strategies for prevention and management is of growing importance. This study aims to analyze global scientific evidence regarding the role of diet in the prevention and management of prostate cancer, while also examining the social, economic, and cultural factors that influence the implementation of these strategies within the Mexican population.

## 1. Introduction

Prostate cancer is the most frequently diagnosed neoplasm in men and represents a critical health concern. In Mexico, it ranks as the second leading cause of death from malignant tumors [1]. Therefore, the implementation of effective prevention strategies is urgently needed. Health promotion has gained increasing relevance in recent years, emphasizing the importance of early preventive action. If not diagnosed early, prostate cancer can metastasize, significantly decreasing long-term survival. In response, Mexico’s Ministry of Health has recommended dietary modifications to reduce the risk of prostate cancer. However, prevention efforts must be tailored to the cultural and demographic characteristics of each population. It is essential to generate and disseminate evidence that resonates with and benefits the Mexican population [2].

The prevalence of prostate cancer continues to rise, mainly due to the strong association between age and disease onset. It is more frequently diagnosed in men over the age of 50, with incidence rates increasing significantly after 65 years of age [1]. Over the last decade, nutrition has emerged as a key area of focus in oncological research, particularly due to the growing recognition of overweight and obesity as risk factors for cancer development. Adipose tissue, closely associated with obesity, plays a pivotal role in prostate cancer pathophysiology [3]. Thus, individuals at high risk should be educated about appropriate nutritional practices, while patients already diagnosed with prostate cancer should receive personalized dietary guidance [4].

This study aims to analyze the global scientific literature on diet as a strategy for prostate cancer prevention and management, with special attention to the social, economic, and cultural factors that influence the implementation of such strategies in the Mexican population.

### Overview of Prostate Cancer in Mexico

Prostate cancer is a significant and persistent public health issue in Mexico. By 2020, cancer had become a national health priority, with prostate cancer identified as one of the main priorities to be addressed. In response, by 2022, governance mechanisms and the national network for integrated cancer care were reinforced. Prostate cancer was the second most common neoplasm diagnosed among Mexican men aged 40 to 80 years [5], accounting for 12.1% of all cancer types within this population. Globally, it ranked fifth in cancer incidence in 2020 [4], and it remains the second most common malignancy in men worldwide, with 1,414,259 new cases, representing 7.3% of all cancer diagnoses globally. These figures underscore the urgency of implementing physical and behavioral interventions to minimize the biological consequences of chronic prostatitis, lower urinary tract symptoms, and prostate cancer among Mexican men [1,6].

Although all men are potentially at risk of developing prostate cancer, current data show that this neoplasm predominantly affects older populations. Screening programs for early detection present specific challenges, often leading to controversial decisions regarding treatment [7]. Available treatments, particularly second-line chemotherapy, may result in severe physical and emotional side effects. Furthermore, despite the national burden of this disease, the scientific evidence available in Mexico to guide clinical decisions is limited or even nonexistent.

Another significant concern is the low percentage of patients who initiate medical care with specialists upon diagnosis of prostate cancer [1]. Healthcare infrastructure varies considerably across Mexican states, leading to disparities in cancer outcomes. These disparities often favor individuals with legal residency or those occupying middle to upper socioeconomic strata, regardless of whether they have health insurance. In contrast, the majority of men do not initially present with severe urinary symptoms, which frequently delays their medical evaluation. This delay contributes to late-stage diagnoses and reduced treatment options.

Due to social customs, some subpopulations may mistrust urological services. Therefore, assessing the need for urology care in Mexico requires a representative sample of the male population to ensure culturally sensitive and accessible interventions [2].

## 2. Context and Current Situation of Prostate Cancer in Mexico

Prostate cancer is one of the most prevalent malignancies worldwide and, in Mexico, mainly affects men over 45 years of age; national surveillance data indicate that new cases are rising by at least 4% annually. Consistent with global trends, prostate cancer is the sixth most common cancer in the country, and its incidence has tripled since 1986. More than half of new diagnoses and over 60% of related deaths occur among men without access to social security services [5]. Incidence and mortality are unevenly distributed across the nation, with certain states reporting rates up to three times higher than the national average. Lifestyle, family history, and pre-existing pathological conditions are major contributors to risk.

The growing prevalence of prostate cancer imposes a considerable economic burden on Mexico’s healthcare system. Analyses covering 2008–2020 show an average age-adjusted incidence of 74.78 per 100,000 men, peaking in the 65–69-year age group. Five-, ten-, and fifteen-year survival rates are estimated at 95.16%, 90.13%, and 84.48%, respectively [8]. Despite ongoing efforts to expand oncology services, there is an urgent need for national prevention strategies that include evidence-based nutritional guidelines and coordinated intersectoral initiatives.

Multiple etiological factors—genetic predisposition, hormonal status, advancing age, and environmental exposures—drive prostate cancer development. Increasing evidence connects diet, metabolic processes, and gene–environment interactions to disease risk. Modifiable factors such as diet, physical activity, and tobacco use, along with broader socioeconomic determinants, markedly influence prostate cancer incidence [9]. Access to healthcare and education further shapes awareness and participation in preventive measures, especially among vulnerable populations.

In Mexico, these vulnerable groups bear a disproportionate burden of modifiable risks and social determinants. Future research should focus on these disparities and develop tailored interventions to lower prostate cancer risk and improve overall health outcomes.

### Social and Economic Impact

Prostate cancer generates a considerable economic burden in Mexico. Approximately 60% of the total budget allocated to this disease is directed through the public health sector, covering direct expenses such as medical treatment, diagnostic tests, procedures, hospitalizations, and complications. Additionally, the condition leads to substantial productivity loss among patients. In Mexico, the estimated annual per capita cost associated with prostate cancer is USD 6806, representing approximately 42% of healthcare expenditures in patients with advanced disease [10]. In early-stage prostate cancer, the cumulative treatment cost for the first year is approximately USD 2300, increasing to USD 4351 in the second year [11].

Interestingly, some reports suggest that the economic burden may be higher in women; however, in men, the repercussions extend beyond financial dimensions. Family members often experience significant emotional strain, not only due to the progression of the disease but also as a result of socioeconomic adjustments. Families may be forced to reduce healthcare spending, and male patients might struggle to fulfill traditional patriarchal roles, leading to psychosocial distress [12].

Comparative analyses of costs across Mexico, Chile, and Brazil indicate that the majority of indirect expenditures stem from work absenteeism. Moreover, the economic impact of prostate cancer is not confined to the individual—it also affects families and broader societal structures. Health is a collective social good, and illness has ripple effects, influencing aspects such as fertility trends, labor force participation, and even migration patterns [13].

Health inequalities are closely associated with elevated morbidity, with marked disparities observed across population groups. In Latin America, the rising incidence of prostate cancer is increasingly linked to preventable risk factors, such as poor dietary habits. Nutritional interventions hold promise for mitigating the disease burden. However, prostate cancer continues to strain Latin American healthcare systems, not only due to its rising incidence among aging populations, but also because of economic implications [7].

Given the growing burden of disease among older adults, it is imperative to promote healthier lifestyles and foster economic resilience within households, particularly in high-risk regions [10]. Marginalized individuals face distinct challenges, including social stigma, misinformation, and fear surrounding prostate cancer diagnosis and treatment. Cultural taboos often inhibit men from openly discussing urological symptoms, and many struggle with disruptions to traditional identities. Concerns regarding sexual function and the emotional toll on partners are frequently voiced, particularly among elderly couples [7].

Consequently, the implementation of personalized emotional, social, and psychological support services is crucial—not only for patients, but also for their families. Health promotion must go hand in hand with psychosocial care. For this reason, recent studies have explored the therapeutic value of certain nutrients and their role in reducing prostate cancer risk and improving patient outcomes, underscoring the potential of nutrition as a complementary strategy in disease prevention and management.

## 3. Role of Nutrition in the Prevention of Prostate Cancer

Recent research highlights the central role of nutrition in prostate cancer prevention and progression [4]. A diet high in processed and animal-based foods—particularly saturated fats, dairy products, industrial trans-fatty acids, excess omega ω-6 fatty acids, phosphatidic acid, calcium, leucine, folic acid, choline, iron, and refined sugars—is associated with an increased risk of developing or accelerating the progression of prostate cancer. Conversely, Mediterranean-type and other plant-forward dietary patterns rich in well-known antioxidant and anti-inflammatory nutrients—as well as foods such as tomatoes, pomegranates, spices, legumes, and a wide variety of vegetables and fruits—have been linked to a lower incidence of prostate cancer and may even enhance chemotherapeutic outcomes in regions with medium-to-high life expectancy [14].

In Mexico, dietary patterns including strictly raw, raw-till-four vegan, and paleolithic-style diets (with moderate carbohydrate intake) have been explored for their potential to reduce prostate cancer risk. Hierarchical linear regression analysis of international incidence data versus daily nutrient intake among Mexican men suggests that higher consumption of animal products is a significant negative predictor—i.e., it is associated with greater risk of prostate cancer onset [14].

Over the past two years, multiple nutritional factors have been investigated from a primary prevention perspective. Although some relationships are well established, many remain preliminary. Notably, nutrition can exert contrasting effects: certain dietary constituents appear protective against prostate cancer, whereas others may promote its initiation and progression. Because modifying dietary habits is relatively simple and cost-effective, individualized recommendations—ideally aligned with a patient’s genetic profile—are warranted [9]. Even seemingly small dietary choices can materially influence cancer risk; as knowledge in this field grows, nutrition is expected to play an increasingly influential role in prostate cancer prevention and management.

### 3.1. Scientific Evidence on Diet and Prostate Cancer

Food processing, preservation, and cooking methods can influence the presence or activity of harmful substances such as pesticides and residual antibiotics. These processes may reduce the carcinogenic potential of certain foods and, thus, play a protective role in reducing cancer incidence. However, few studies have thoroughly addressed this aspect in the context of prostate cancer [15]. While evidence supporting an association between diet and cancer—particularly prostate cancer—remains inconclusive, existing dietary recommendations for cancer prevention emphasize limiting the consumption of energy-dense foods with low nutritional value, as they contribute to overweight and obesity, both recognized risk factors for various types of cancer [16].

Population-based studies investigating the relationship between dietary patterns and cancer incidence have yielded mixed results. Nonetheless, adherence to dietary models such as the Mediterranean, Atlantic, or French diets—characterized by a high intake of fruits, vegetables, seeds, fish, and healthy oils—has been associated with a reduced risk of prostate cancer. Similarly, a protective effect has been suggested for vegetarian diets in some studies. However, this finding was contradicted in a specific study conducted among Mexican men in the state of Querétaro, where vegetarian dietary patterns were associated with an increased risk of prostate cancer [1].

Observational studies are critical for generating hypotheses about potential causal relationships, but their findings should be interpreted with caution. These studies often involve populations that are not fully representative of the global population, and their results may vary significantly depending on sociodemographic, economic, genetic, health-related, and behavioral factors among diverse human subgroups [4]. Current research evaluating the effects of diet or dietary biomarkers on prostate cancer remains limited and preliminary. Longitudinal studies assessing both dietary intake and validated biomarkers are essential to establish causality and determine the specific impact of dietary components on disease outcomes.

### 3.2. Beneficial Foods and Nutrients

Several foods and nutrients have demonstrated potential benefits in reducing the risk of prostate cancer, primarily due to their antioxidant, anti-inflammatory, and immunomodulatory properties. These protective compounds are especially abundant in fruits and vegetables, including carotenoids, polyphenols, isoflavones, and various vitamins and minerals. Whole grains, nuts, seeds, and fatty fish rich in omega-3 fatty acids also contribute significantly to dietary protection [14].

Fruits and vegetables are particularly valuable due to their high content of fiber, vitamins, and bioactive phytochemicals. Among the most protective compounds identified are sulfur-containing compounds, polyphenols, flavonoids, lignans, soluble fiber, vitamin E, and selenium [17]. These nutrients collectively help reduce oxidative stress and inflammation, two key mechanisms involved in cancer development.

Tomato-derived products, especially those with enhanced lycopene bioavailability, such as tomato paste or cooked tomatoes, have been consistently associated with a reduced incidence of prostate cancer. Both dietary intake and circulating levels of lycopene have shown inverse associations with prostate cancer risk [18]. Additionally, soy-based foods—particularly in populations with high habitual soy consumption—are associated with a lower incidence of the disease, likely due to the presence of phytoestrogens and isoflavones.

Among the compounds most strongly implicated in prostate cancer protection, many occur in high concentrations in whole-grain foods. These grains provide beneficial fatty acids, lignans, vitamin E, zinc, and selenium, and—though to a lesser extent—phytoestrogens (Figure 1). These nutrients often act synergistically, and consuming them in their natural food matrix is preferable to isolated supplementation. Regularly incorporating whole grains, seeds, and other plant-based sources into a varied and balanced diet not only supports overall health but also raises public awareness of nutrition’s preventive role in prostate cancer control [19]. Consistent with this view, multiple studies report a moderate-to-high inverse association between whole-grain intake and prostate cancer. The array of bioactive compounds present in these grains exhibits anticancer activity both individually and through synergistic interactions with other nutrients and dietary fiber [20].

### 3.3. Harmful Foods and Nutrients

Saturated fats have long been studied as potentially harmful nutrients that may contribute to the development of prostate cancer. Certain unhealthy foods should be considered dietary agents that increase the risk of prostate cancer development [21].

a. Red meat intake. Red meats are high in iron and may elevate the risk of developing prostate cancer, especially when cooked at high temperatures [22]. Public health recommendations should discourage red meat consumption as part of preventive strategies.

b. Processed meats. Inadequate cooking methods can potentially promote carcinogenesis. Processed meats or those containing preservatives increase cancer risk, primarily because they can lead to the formation of *N*-nitroso compounds. Additionally, some are smoked using hydrocarbon smoke. These meats should be replaced with healthier alternatives such as preservative-free meats [23]. Cooking methods must follow recommended temperature and time guidelines, favoring methods that reduce carcinogen formation.

c. Refined sugars. While the evidence linking refined sugars to prostate cancer is not conclusive, limiting their intake is advised due to their association with other chronic diseases such as diabetes and cardiovascular conditions [24]. When consumed, sugars should remain within quantities aligned with a healthy, balanced diet, adjusted according to an individual’s physical activity level.

In developing countries, it is especially critical to avoid processed meats, as compounds such as sodium nitrate and sodium nitrite—used to limit *Clostridium botulinum* growth and enhance flavor and color—can produce *N*-nitroso compounds in the stomach [15] (Figure 1). Ultimately, men should avoid processed meats, opt for preservative-free alternatives, and ensure proper cooking techniques that minimize browning and the formation of harmful compounds.

## 4. Ideal Diet for Prostate Cancer Patients in Mexico

Interventions aimed at the management and prevention of prostate cancer should consider not only scientific evidence but also the sociocultural and economic context of the target population. A well-crafted dietary recommendation that incorporates local flavors, cultural practices, and affordability is more likely to be adopted, making it a key strategy in addressing this disease. Prostate cancer remains the second most frequent type of cancer among Mexican men. Therefore, dietary guidelines should emphasize complete meals rather than isolated recipes, allowing individuals to choose culturally appropriate foods that meet nutritional criteria based on regional customs and accessibility [13].

Several studies have highlighted the protective effect of various foods produced in Mexico, many of which are deeply embedded in the country’s gastronomic tradition. Integrating these local foods—such as cereals, vegetables, legumes, seeds, and herbs—into national dietary guidelines should be prioritized, taking into account the diverse regional contexts across the country.

The overarching goal is to improve diet and nutrition for cancer prevention, treatment, control, and survivorship, ultimately promoting the adoption of long-term healthy lifestyles that reduce the risk of chronic diseases. An ideal diet should adequately compensate for an individual’s energy expenditure while offering balanced nutritional intake [4]. Key food groups in a healthy eating pattern include fruits and vegetables; cereals, grains, and tubers; legumes; healthy oils and fats; and foods of animal origin, each fulfilling essential biological functions [25].

These “healthy plate” proportions can be tailored to meet specific dietary and caloric requirements. In the context of prostate cancer, particular emphasis should be placed on functional foods rich in potent antioxidants with anticancer properties, such as those found in vegetables, fruits, and dairy products [24]. In cases where caloric needs are not met, energy-dense foods can be included to ensure adequate intake.

### 4.1. Cultural and Socioeconomic Considerations

Culturally, food plays a fundamental role in social traditions and identity, which can significantly influence the acceptance and adherence to dietary recommendations. Factors such as lifestyle, religious practices, regional customs, and sociocultural context shape individual dietary habits. Evidence shows that adherence to nutritional guidelines improves when recommendations align with familiar cultural patterns and personal taste preferences. In Mexico, dietary practices vary considerably by region. From a traditional breakfast of chilaquiles with fresh orange juice in Mexico City, to barbacoa in Hidalgo, ceviche in Guerrero, fish tacos in Baja California, sopa de lima in Yucatán, and a wide array of regional sweets and quesadillas throughout the country, the diversity of the national diet must be acknowledged and respected in dietary planning [13].

Given the cultural importance of food and mealtimes, dietary recommendations that conflict with established traditions may have a significant impact within households, potentially reducing adherence. Unlike in parts of Europe and North America, where “food deserts” limit access to fresh produce and promote dependence on processed foods, most urbanized regions of Mexico—particularly in the north—have reasonable access to fresh ingredients. In this context, dietary choices are often more strongly influenced by cultural traditions than by geographic or socioeconomic constraints. However, at the household level, socioeconomic status still plays a pivotal role. It can determine a family’s ability to purchase fresh produce, attend follow-up nutrition consultations, and implement meal planning strategies [11].

Socioeconomic status may also influence health literacy. While this is not universally applicable, lower income levels are frequently associated with limited educational attainment, which can hinder comprehension and adherence to nutritional advice. Therefore, any dietary recommendation must take into account the patient’s socioeconomic context to ensure practical applicability and effectiveness.

In clinical practice, minor hemorrhage following prostate biopsy is common, underscoring the need to review and potentially discontinue anticoagulant medications before the procedure to minimize bleeding risk. Beyond the medical preparation, it is essential to evaluate the patient’s current dietary patterns. This can be facilitated by discussing recent home-cooked meals, exploring lifestyle factors such as current and ideal body weight, physical activity levels, and the patient’s readiness to make dietary changes. While many patients may already consume foods aligned with recommended low-fat, low-calorie dietary patterns, a more nuanced and individualized discussion is often necessary to support behavior change [16].

### 4.2. Accessible and Affordable Foods

As in many predominantly low-income regions around the world, a significant number of men in Mexico lack access to affordable, nutritionally adequate foods that could help prevent or manage prostate cancer. According to recent reports, the second most common barrier to food access is the lack of financial resources, surpassed only by rising food prices. Specifically, fresh fruits and vegetables, high-quality proteins, and diverse, balanced meals were identified as the most difficult to include in the diets of populations facing poverty and food insecurity. Other contributing factors include the high cost of these items, lack of appetite, and reduced energy and nutrient requirements, especially in older adults [13].

Various dietary strategies have been reported to help reduce the cost of accessing nutritious foods. These include efforts to improve access to fruits, vegetables, and whole grains. One approach involves diversifying household income sources and decreasing dietary monotony among low-income populations—for instance, encouraging greater participation in the family economy and promoting the inclusion of fruits and vegetables to reduce dependence on maize as the primary dietary staple [24].

Another practical and sustainable strategy involves increasing food self-sufficiency through community-based initiatives. These include the development of home vegetable gardens and the promotion of non-polluting domestic agriculture. In rural and marginalized areas, these initiatives are particularly relevant, as they aim to reduce food costs while improving the availability of recommended foods [20].

To be effective, low-cost dietary models must align as closely as possible with cultural preferences and traditional food practices. They should incorporate ingredients that are recognizable and readily available in local community markets, thereby increasing the likelihood of adherence. These models must also be adaptable to regional availability and affordability constraints while promoting a more diversified, nutrient-rich diet.

### 4.3. Portion and Frequency Recommendations

Portion recommendations for each meal, based on the distribution of macronutrients, should be tailored to the patient’s age, physical activity level, and individual nutritional assessment. The proposed dietary plan is structured around five meals per day, with macronutrients incorporated primarily into the main meals. While an afternoon snack is not included by default, it can be added according to cultural practices or increased physical activity demands.

The plan includes three main meals—breakfast, lunch, and dinner—each of which should contain a balanced combination of low-fat proteins, healthy fats, and varied carbohydrate sources. Every primary meal should incorporate fruits and/or vegetables, either as a side dish or as part of the main course. Between meals, snacks should feature healthy foods drawn from all food groups and contribute to daily nutritional requirements [22]. Adequate hydration is also essential and should be emphasized throughout the day.

In Mexico, afternoon snacks (“merienda”) are a common dietary habit and often include nutrient-rich foods. A culturally adapted list of suggested snacks may be provided to enhance dietary adherence while respecting local customs.

Meal timing is an important component of dietary planning. Properly spaced meals support the health and functionality of both the urinary and digestive systems. Therefore, guidance on optimal timing for each meal is included, encouraging consistent patterns that align with the body’s natural rhythms and improve overall well-being.

## 5. Specific Foods with Potential Benefits

Tomatoes (fresh and processed): Tomato consumption has been consistently associated with a reduced risk of prostate cancer. This protective effect is attributed to bioactive compounds such as lycopene, a carotenoid with potent antioxidant properties, as well as ascorbic acid (vitamin C), polyphenols, flavonoids, and vitamin E. In addition to their nutritional value, tomatoes hold cultural significance in Mexico, where they are a staple in everyday cooking and festive dishes. Their widespread acceptance facilitates their integration into dietary interventions aimed at promoting healthier eating habits.

Avocado: Mexico leads the world in per capita consumption, with an estimated 10.5 kg per person in 2018. Avocados are an excellent source of dietary fiber, which contributes to maintaining a healthy weight—an important factor in the prevention of prostate cancer [16]. They also provide monounsaturated fats and antioxidants. The recommended portion is approximately 10 to 15 g per main meal, consumed in moderation to ensure nutritional balance. Avocados are already well integrated into traditional Mexican meals, commonly served with tacos, sandwiches, and refried beans, which enhances their acceptability in health promotion strategies [26].

Fruits and Vegetables: The promotion of one or more daily servings of fruits and vegetables is crucial, as these foods provide essential vitamins, minerals, fiber, and phytochemicals with antioxidant, anti-inflammatory, antiproliferative, and antiangiogenic properties. These compounds contribute to cancer risk reduction and may improve overall health and quality of life. For vegetables, the maximum recommended cooking temperature is approximately 100 °C to preserve nutritional value. In the case of tomatoes, cooking is beneficial, as it increases the availability of trans-lycopene, the biologically active form. Additionally, cooking helps reduce antinutritional factors, enhancing overall bioavailability of protective nutrients [14].

### 5.1. Tomatoes and Lycopene

Tomatoes are rich in essential nutrients, including vitamins C, E, and K; carotenoids such as lutein, alpha-carotene, and beta-carotene; and flavonoids like naringenin, quercetin, apigenin, and kaempferol [18]. In Mexico, tomatoes are widely consumed and constitute the basis of various sauces, including adobos and moles, which are integral to the country’s traditional cuisine. Notably, the most bioavailable forms of lycopene, a key compound associated with reduced prostate cancer risk, are found in cooked and processed tomato products such as tomato paste, tomato sauce, and ketchup—all common elements in the Mexican diet. The integration of these foods into nutritional strategies represents a culturally relevant and practical approach to prostate cancer prevention [27].

Tomatoes and tomato-based products contain an array of phytochemicals, including carotenoids, polyphenols, and nutrients such as vitamins C and E, and potassium [27]. The red pigments and beta-carotenoids in tomatoes provide antioxidant effects that contribute to their protective properties [28]. Lycopene tends to accumulate in the prostate, where it interacts with tissues and demonstrates low systemic levels, with some studies noting a PSA reduction following lycopene supplementation [29]. It is worth noting that only moderate amounts of tomatoes are needed to observe potential health benefits. An average intake of 160 g per week may suffice to positively influence prostate cancer risk [30,31] (see Figure 2).

Lycopene is readily available in tomato-based products, which may contribute to the prevention of prostate cancer, particularly when integrated early into dietary interventions. The association between lycopene and tomatoes lies in the compound’s bioavailability and its ability to penetrate prostate tissue, emphasizing the importance of leveraging tomato-derived products to enhance their protective role against this condition [27].

The most significant biological property of lycopene—and the one most often cited in the literature—is its strong antioxidant capacity. Lycopene is a non-provitamin A carotenoid; unlike other carotenoids, it cannot be converted into vitamin A in the human body. Nevertheless, it protects cells from oxidative damage and may reduce cancer risk [32]. Both intervention and prospective studies continue to evaluate how lycopene bioavailability and different treatment regimens influence prostate cancer incidence [18].

Beyond its antioxidant activity, lycopene exhibits marked anti-inflammatory effects that are highly relevant to prostate health. Because lycopene preferentially accumulates in prostate tissue, it is an ideal candidate for targeted nutritional intervention. Selenium, an essential trace element and co-factor for glutathione peroxidase, provides complementary antioxidant protection via distinct enzymatic pathways. The synergistic potential of lycopene and selenium lies in their combined ability to mitigate oxidative stress through multiple mechanisms while modulating inflammatory cascades that contribute to both benign prostatic hyperplasia (BPH) progression and carcinogenesis.

Consequently, the lycopene–selenium combination represents a promising nutraceutical strategy for managing BPH and potentially lowering prostate cancer risk. By addressing several pathophysiological pathways simultaneously, this pairing offers a more comprehensive therapeutic approach than either compound alone. Research published in Current Medicinal Chemistry has highlighted the enhanced prostate protection achieved by combining antioxidant agents, suggesting that lycopene–selenium interactions are truly synergistic, not merely additive [33].

Clinical evidence documented in the Archives of Italian Urology and Andrology supports nutraceutical approaches for BPH, demonstrating reduced inflammatory markers, improved urinary symptoms, and potential modulation of prostate size when lycopene and selenium are integrated into treatment protocols [34]. A comprehensive review in Nutrients further underscores the growing body of evidence favoring nutraceutical interventions; it positions the lycopene–selenium combination as part of a shift toward precision nutrition strategies that target specific molecular pathways in prostate disorders [35].

This combination may be particularly valuable as a preventive measure for men at elevated risk of BPH or prostate cancer, and as an adjunct therapy for those already diagnosed with benign prostate conditions. Given the favorable safety profiles and documented biological activities of both compounds, this nutraceutical pairing is well suited for long-term management.

Importantly, the therapeutic potential of lycopene and selenium extends beyond symptom control to encompass disease prevention and delayed progression. By addressing fundamental cellular processes—namely, oxidative stress, inflammation, and dysregulated cellular proliferation—this approach offers a holistic strategy for preserving prostate health throughout the aging process. Future research should focus on optimizing dosage regimens, identifying biomarkers of treatment response, and establishing clear clinical guidelines for the implementation of this promising nutraceutical combination.

### 5.2. Avocado and Healthy Fats

Avocado is a fruit rich in healthy fats, primarily monounsaturated fatty acids (MUFAs) such as oleic acid, and also contains various anticancer nutrients, including vitamins A, C, and E, polyphenols, carotenoids, and glutathione [36]. These bioactive compounds have been shown to reduce low-density lipoprotein (LDL) cholesterol and increase high-density lipoprotein (HDL) cholesterol, thereby providing energy while supporting immune and central nervous system health. Additionally, the combination of monounsaturated fats and phytochemicals present in avocados has been associated with a reduced risk of chronic diseases, including gastrointestinal, respiratory, kidney, cardiovascular, and neurodegenerative disorders such as Alzheimer’s disease, as well as certain cancers, including prostate cancer [37].

Epidemiological studies focused on general nutrition have identified an inverse association between monounsaturated fat intake and lethal prostate cancer. These beneficial compounds have also been linked to improved prognosis due to their anti-inflammatory, pro-apoptotic, and antioxidant properties [38].

Among patients with metabolic syndrome, substituting saturated fats with MUFAs has been associated with modest reductions in body fat, even without intentional caloric restriction. Additional benefits include improvements in fasting blood glucose and insulin sensitivity, reduced levels of C-reactive protein, and increased adiponectin concentrations. In a clinical trial involving prostate cancer survivors, daily avocado consumption (175 g/day for men) was associated with a reduction in white blood cell count—an indicator of systemic inflammation [36,39].

Furthermore, one study found that murine macrophages stimulated with lipopolysaccharides and treated with avocado seed extracts for 24 h showed significantly decreased levels of pro-inflammatory cytokines, including IL-6, IL-1β, and tumor necrosis factor-alpha (TNF-α) [40]. Regarding pro-apoptotic effects, avocados contain selenium metabolites that can slow down the cell cycle and inhibit protein synthesis, thereby reducing cell proliferation—particularly in androgen-dependent prostate cancer cells [41,42]. These effects are supported by improved antioxidant capacity in both blood and prostate tissue, suggesting a potential reduction in disease mortality. The antioxidant potential of avocados contributes to neutralizing free radicals—major sources of reactive oxygen species—thereby helping to prevent cellular damage and genetic mutations [41].

In addition to its health benefits, avocado consumption is steadily increasing in Mexico, driven by rising production levels. Given that the distal compounds in avocados are well tolerated by the gastrointestinal system, it is advisable to include them in any meal. Preliminary research suggests that adding avocados to salads, sauces, or breads may enhance the availability of beneficial compounds. Avocado’s culinary versatility also makes it ideal for integration into diverse dishes. Beyond traditional guacamole, it can be consumed in savory preparations, desserts, and soups [39]. However, further research is warranted to evaluate the specific benefits of avocado consumption within Mexican culinary patterns, especially in the context of prostate cancer prevention and management.

### 5.3. Antioxidant-Rich Fruits and Vegetables

Beyond the role of fat consumption, the inclusion of fresh, antioxidant-rich fruits and vegetables in the diet has been extensively highlighted as a functional strategy for cancer prevention. Antioxidants are known to reduce oxidative stress, help prevent atherosclerosis, lower the risk of cancer and cardiovascular diseases, and exert anti-inflammatory effects. These naturally occurring compounds—such as vitamins A, C, and E, selenium, phenolics, and polyphenols—can counteract the harmful impact of reactive oxygen species (ROS) on cellular structures [20]. Antioxidants influence key cellular signaling pathways, prevent damage caused by free radicals, promote the activity of detoxification enzymes, and inhibit the function of ROS-generating enzymes [19].

Some antioxidants, including vitamin C and polyphenolic compounds such as flavonoids and isoflavonoids, have demonstrated the potential to modify the prostate gland microenvironment and inhibit carcinogenesis in *in vitro* models [20]. These findings are further supported by *in vivo* studies showing the protective effects of antioxidants and other phytochemicals on prostate tissue. For instance, vitamin C—abundant in fruits and vegetables—has been shown in experimental studies involving rodent models to reduce prostate hyperplasia and the incidence of benign prostate hyperplasia [14].

Fruits and vegetables are among the richest dietary sources of antioxidants, containing not only vitamin C and carotenoids but also various bioactive phytochemicals with potential anti-carcinogenic properties. These components contribute significantly to the prevention of chronic degenerative diseases, particularly cancer. Polyphenolic compounds, including flavonoids, are especially notable for their biological effects in modulating inflammation and cellular proliferation.

A variety of foods with demonstrated health benefits have been consistently characterized as part of diets rich in colorful fruits and vegetables, which offer a diverse array of bioactive phytochemicals [43]. These foods reflect the refinement of nutritional advantages shaped through natural selection and provide essential compounds that contribute to cellular protection and overall health. Therefore, the inclusion of antioxidant-rich and phytochemical-dense fruits and vegetables should be promoted not only as part of a preventive health strategy but also as a viable, culturally relevant component of local food markets. Their integration represents a significant opportunity to enhance population health through accessible, affordable, and culturally accepted dietary choices.

### 5.4. Whole Grains and Fiber

Fiber, a non-digestible component of plant foods, is one of the elements that constitute healthy eating for disease prevention and overall well-being. One of its proposed benefits is the reduction in cancer risk. It modulates the gut microbiota, providing several preventive effects by contributing to the production of short-chain fatty acids (SCFAs) with anti-inflammatory properties, forming barriers against the proliferation of pathogenic microorganisms, and decreasing inflammation not only in the intestinal tract but also in other organs with which it interacts [44]. Additionally, fiber reduces the transit time of intestinal contents, limiting their contact with the colon and thereby reducing the reabsorption of secondary bile metabolites, some of which may be carcinogenic [37]. It is important to note that fiber should be consumed in its natural form, as supplementation has not been associated with the same protective effects.

In the Mexican population, most fiber intake comes from starchy foods, with only about 7% derived from fruits and vegetables. One strategy to increase fiber consumption is through the intake of whole grains such as quinoa, brown rice, rye, oats, millet, and barley, among others. It is advisable to consume at least three servings per day (Figure 2), although their consumption in Mexico—as in many other countries—remains low. In addition to fiber, these grains are sources of vitamins and minerals that contribute to general health and well-being [15]. In Mexico, the most accessible and widely grown whole grains are quinoa, brown rice, and oats. These grains do not have to be served only as side dishes. Quinoa functions as a legume, grain, and complete vegetarian protein all in one. As such, it can aid digestion, lower blood sugar and cholesterol levels, and reduce the risk of heart disease. Its soluble fiber content can also help decrease abdominal fat. Brown rice is considered healthier than white rice due to its higher fiber and protein content. It is low in calories and sugar, making it suitable for people with diabetes. Additionally, it may help with weight management by promoting satiety and reducing excessive eating. Oats are rich in fiber, heart-healthy vitamins and minerals, and a mix of antioxidants [45]. They are also nutritionally dense, providing higher amounts of protein. It is recommended to incorporate at least one of these grains into each main meal of the day (Figure 2).

## 6. The Gut Microbiome–Prostate Cancer Axis: Mechanisms and Nutritional Modulation

The emerging concept of the gut–prostate axis has revolutionized our understanding of prostate cancer biology, revealing a complex bidirectional communication pathway between the intestinal microbiota and prostatic health. Recent investigations have shown that the gut microbiome is significantly influenced by various environmental factors, particularly lifestyle modifications, and that alterations in its composition may directly contribute to prostate cancer progression through bacterial metabolites and endotoxins [46,47].

This relationship is mediated through both direct and indirect mechanisms, with microbes or their products impacting the prostate either locally—within the prostate and urinary tract—or systemically through signals originating in the gastrointestinal microbial communities [48]. Understanding these intricate molecular mechanisms underlying the gut–prostate axis is essential for developing targeted therapeutic interventions that leverage microbiome modulation for prostate cancer prevention and treatment.

Specific bacterial populations have been consistently associated with prostate cancer risk and progression, with certain strains demonstrating distinct pathogenic potential. Studies have identified *Bacteroides massiliensis*, *Bacteroides*, and *Streptococcus tissierellaceae* as being associated with an increased risk of prostate cancer development, although the exact mechanisms by which these microbes contribute to carcinogenesis remain to be fully elucidated [47]. More notably, men with castration-resistant prostate cancer exhibit a higher abundance of gut bacteria with androgenic functions, suggesting that microbial communities actively participate in androgen metabolism and hormone-dependent cancer progression [49].

This microbial involvement in steroid hormone metabolism represents a novel pathway through which the gut microbiome may influence prostate cancer biology, particularly in the context of treatment resistance. Furthermore, men with high-risk prostate cancer share a distinct gut microbial profile, indicating that microbiome profiling could serve as an effective screening tool for identifying patients with aggressive disease phenotypes [49]. The diagnostic and prognostic utility of microbiome analysis in prostate cancer extends beyond bacterial identification to include functional assessments of microbial communities. Recent research has demonstrated a strong association between urinary and gut microbiome composition and both the incidence and progression of prostate cancer, with the urinary microbiome emerging as a promising non-invasive biomarker for early detection and risk assessment [50].

Altered microbial profiles in prostate cancer patients represent a significant advancement in non-invasive diagnostics, offering the possibility of identifying high-risk individuals before clinical symptoms appear. Moreover, reduced fecal microbiota alpha-diversity has been correlated with greater prostate tumor burden, suggesting that the overall health of the microbial ecosystem may serve as an indicator of disease progression [51].

The mechanistic relationship between gut microbiota and prostate cancer involves complex metabolic interactions that directly influence tumor growth and progression. Gut bacteria produce a variety of metabolites that can either promote or inhibit carcinogenesis. Notably, short-chain fatty acids (SCFAs), formed through bacterial fermentation of plant-based fibers, exhibit protective effects against cancer development [52]. In contrast, other bacterial metabolites and endotoxins can induce a pro-inflammatory environment that favors tumor growth, highlighting the dual nature of microbial influence on cancer biology [46].

Additionally, the gut microbiota significantly affects drug metabolism and toxicity, representing a critical mechanism through which bacteria can modulate cancer treatment outcomes. These microbial activities may enhance therapeutic response by improving drug bioavailability or, conversely, reduce efficacy due to drug deactivation or increased side effects [47]. These findings underscore the importance of maintaining a balanced microbial ecosystem to optimize both cancer prevention and treatment efficacy.

Nutritional interventions represent a promising and clinically feasible approach for modulating gut microbiota composition to reduce prostate cancer risk and improve treatment outcomes. Omega-3 polyunsaturated fatty acid (PUFA) supplementation has demonstrated significant efficacy in slowing prostate cancer progression. This effect is partly attributed to the reduction of gut *Ruminococcaceae* populations and the associated decrease in fecal butyrate levels [51]. This intervention specifically targets the microbiome–cancer interface by altering microbial populations that contribute to tumor growth, while simultaneously decreasing inflammatory metabolites that promote carcinogenesis.

The use of omega-3 fatty acids thus represents a precision nutrition strategy that confers both direct anticancer benefits and indirect microbiome-mediated advantages. This dual mechanism highlights their potential in cancer prevention and as adjuvants in therapeutic protocols.

Prebiotic supplementation offers another targeted approach for beneficial modulation of the gut microbiota in the prevention and management of prostate cancer. Prebiotics are generally non-digestible compounds that selectively promote the growth of beneficial bacterial strains and have demonstrated anti-inflammatory and anticancer properties [50]. These compounds function by serving as substrates for protective microbial populations while simultaneously creating an intestinal environment that is less favorable to pathogenic bacteria associated with tumor progression.

Due to their selective nature, prebiotics allow for precise microbiome engineering, enhancing protective bacterial functions and suppressing cancer-promoting microbial activities. Current evidence suggests that sustained prebiotic intake can modify gut microbiota composition and reduce prostate cancer risk by fostering beneficial microbial communities [47].

Probiotic interventions offer a direct method of restoring microbial balance and enhancing anticancer immune responses in patients with prostate cancer. Specific strains of *Lactobacillus* and *Bifidobacterium* have demonstrated measurable improvements in microbiota composition. These beneficial bacteria produce metabolites that actively inhibit tumor cell proliferation and strengthen immune surveillance mechanisms [52].

Probiotics exert their effects through multiple pathways, including the competitive exclusion of pathogenic bacteria, production of antimicrobial compounds, enhancement of intestinal barrier integrity, and modulation of both local and systemic immune responses. Implementing probiotic-based strategies may help prevent the onset of prostate cancer by reestablishing a healthy microbial balance that supports antitumor immunity [46]. Furthermore, combining probiotics with conventional cancer treatments has the potential to enhance therapeutic efficacy and reduce treatment-related complications.

Dietary fiber intake represents a fundamental nutritional strategy for modulating the gut microbiota, directly impacting prostate cancer risk through multiple mechanisms. High-fiber diets promote the proliferation of beneficial bacteria that produce short-chain fatty acids (SCFAs), particularly butyrate, which exhibits notable anticancer properties, such as promoting apoptosis of cancer cells, inhibiting tumor angiogenesis, and enhancing immune cell function [52].

The fermentation of dietary fiber by gut bacteria creates an intestinal environment less conducive to the growth of pathogenic microbes, while simultaneously generating metabolites that strengthen anticancer immunity. Plant-based dietary patterns, naturally high in prebiotic fibers, have been consistently associated with reduced risk of prostate cancer and improved treatment outcomes. These findings support the role of fiber-driven microbiome modulation as a cornerstone of nutritional strategies for cancer prevention.

Advanced therapeutic approaches involving fecal microbiota transplantation (FMT) represent the frontier of microbiome-based interventions for prostate cancer. FMT consists of transferring healthy gut microbiota from rigorously screened donors to patients, with the goal of rapidly restoring a balanced microbial ecosystem and correcting dysbiosis associated with cancer progression [50].

This strategy has shown particular promise in enhancing the efficacy of cancer immunotherapy by reestablishing microbial populations that support robust anti-tumor immune responses. Targeting the gut microbiota through interventions such as prebiotics, probiotics, or FMT may increase the effectiveness of androgen deprivation therapy and other standard prostate cancer treatments. By optimizing the microbial environment, these strategies may improve treatment responsiveness and reduce adverse effects [12].

The integration of microbiome restoration with conventional cancer therapies represents a paradigm shift toward precision medicine—one that considers the patient’s microbial ecosystem as a critical factor in treatment planning and clinical decision-making.

The clinical translation of microbiome research into practical prostate cancer management requires the development of personalized approaches that take into account individual microbial profiles, genetic susceptibility, and treatment history. Future therapeutic strategies will likely involve comprehensive microbiome profiling to identify patients who would benefit the most from specific microbial interventions.

These strategies could then be followed by targeted nutritional or probiotic treatments designed to optimize the gut–prostate axis, either for cancer prevention or as adjunctive therapy. Integrating microbiome analysis with traditional cancer biomarkers may enhance risk stratification, refine treatment selection, and ultimately improve patient outcomes [50].

This precision medicine approach acknowledges that effective prostate cancer management must address not only the tumor itself but also the complex microbial ecosystem that modulates cancer biology, influences therapeutic response, and impacts long-term survival.

## 7. Challenges and Limitations in Implementing an Ideal Diet

In an ideal scenario, a dietary strategy aimed at preventing prostate cancer and mitigating the adverse effects of cancer treatment would promote the consumption of fruits, vegetables, and whole grains, while limiting the intake of animal fat, saturated and trans fats, salt, and sugar-sweetened beverages [19]. However, in Mexico, many communities live in food deserts where access to healthy food is severely limited. Similar to other countries with pronounced economic disparities, obesity has become an epidemic in lower-income communities with limited prospects for change. Additionally, while there is increasing advocacy for consuming local, fresh, low-emission foods, implementing these practices poses a significant challenge under current socioeconomic conditions. The average monthly gross domestic product per household is approximately USD 520, a limited amount that constrains the ability to plan for and purchase nutritionally adequate foods [17].

Therefore, promoting warranted dietary interventions—such as reducing sugar consumption—is essential. Nutrition and healthcare should be recognized as key components of community-based programs, such as those implemented by Portland Community College, which emphasize food education and access [53]. Given the low consumption of fruits and vegetables in many at-risk groups, it is crucial to direct nutrition intervention efforts where they can effectively address barriers and structural problems.

Fruits and vegetables are often prohibitively expensive, and poor nutrient preservation is a widespread issue, especially in lower-income settings. According to the evidence, two primary intervention approaches are recommended: increasing actual dietary intake and expanding the dissemination of information to raise nutritional awareness. Community-level mechanisms—such as those implemented by local organizations—have demonstrated success in overcoming access barriers and changing behaviors [17].

Public health programs and even pharmaceutical companies have successfully employed strategic advertising to promote the adoption of new dietary practices. These models could serve as effective channels for disseminating dietary guidelines and promoting affordable, nutritious foods as part of a comprehensive cancer prevention strategy.

### 7.1. Access to Healthy Foods in Marginalized Areas

One of the critical determinants of healthy eating is access to nutritious foods; however, this remains a significant challenge, particularly in marginalized areas of Mexico. In these regions, access is often restricted, limited, or inadequate due to a range of factors, including low purchasing power, socioeconomic disparities, geographic isolation, limited education, and sociocultural barriers [24]. These conditions contribute to unhealthy dietary habits, as low-income retail environments typically offer foods high in calories, unhealthy fats, and added sugars—commonly referred to as “food swamps.” Geographically, neighborhoods lacking access to fresh produce and healthy food options are classified as “food deserts”. In these environments, patients are more likely to consume fast food and ultra-processed products, resulting in diets deficient in fruits, vegetables, and fiber. Such dietary patterns, combined with sedentary lifestyles and high obesity prevalence, are associated with poorer cancer outcomes, delayed recovery, and increased risk of treatment-related side effects [4].

Nevertheless, increasing access to healthy foods in underserved Mexican communities is achievable through multifaceted strategies that not only improve physical access to food but also promote education in healthy cooking and meal preparation. Several small-scale initiatives are actively working to improve the availability and affordability of fresh, locally grown produce in these areas. Examples include subsidized community gardens, cooperatively cultivated surplus fruits and vegetables, urban orchards managed by local residents, and commercial food cooperatives. Other strategies involve fair-trade vegetable delivery services, member-based organic food cooperatives, Community Supported Agriculture (CSA) schemes with paid memberships, subsidized farmers’ markets, and local associations promoting organic fruits and vegetables [4].

Some of these efforts are part of broader initiatives that combine health promotion with the development of sustainable local food systems. In low-resource settings, local governments and non-governmental organizations have also begun providing training in small-scale agriculture and food production as a way to enhance food security. For instance, in a peri-urban community of resettled squatters, a comprehensive development program that included training in crop cultivation and small animal husbandry resulted in cost savings and contributed to the reduction of childhood acute malnutrition. In other cases, often outside Mexico, residents have taken it upon themselves to improve access to healthy foods by creating informal produce stands or community-run food services in areas abandoned by supermarkets or grocers [54].

These examples suggest that meaningful collaboration with local communities can help overcome economic and structural barriers to accessing nutritious food. Supporting and scaling such community-based models may be key to improving dietary health and preventing chronic diseases such as prostate cancer in Mexico’s most vulnerable populations.

### 7.2. Cost and Availability of Food

In the context of improving health behaviors for the prevention and management of prostate cancer, the cost of food and its availability are critical issues. Poor diet is a major risk for prostate cancer in Mexico, and it is essential to evaluate the economic, cultural, and social barriers that hinder the improvement of dietary quality [20]. In many Latin American cities, healthy foods such as fruits and vegetables are disproportionately expensive relative to daily income when compared to unhealthy processed products. Specifically, in Mexico, the cost of a healthy food basket can amount to nearly four times the daily minimum wage [22]. This presents a significant burden for low-income families, who would have to allocate a substantial portion of their earnings solely to acquire nutritious foods. For individuals and families living below the poverty line, adhering to dietary recommendations to prevent prostate cancer is simply not feasible. Notably, the high cost of healthy diets is the fourth leading cause of food inequality in the country, with the highest impact observed in states such as Oaxaca, Guerrero, and Chiapas [55].

Another major barrier is the limited availability of local and seasonal food. In Guadalajara, for instance, the availability and prices of local fruits and vegetables vary significantly depending on the season or month, making it difficult to regularly purchase the quantity of fresh produce recommended by nutritional guidelines. When these recommendations are followed, the weekly food budget is often exhausted on produce alone. Compounding this challenge is the short shelf life of many fruits and vegetables, which are only sold once a month at itinerant markets, limiting consistent access. Organic foods are even less accessible. Although no comprehensive study exists, direct observation of three local markets near a specific organization’s office found that only two stalls offered organic fruits and vegetables [22]. Furthermore, there is little to no information available about organic food pricing and seasonal variations, leaving significant data gaps that hinder policy development.

A common perception, especially among lower-income populations, is that a healthy diet is inherently expensive and impractical. This belief is widespread not only in Mexico but also in other countries, including the United States, where many low-income urban residents report difficulties identifying and purchasing affordable, healthy foods. Designing practical, effective interventions requires an understanding of what families perceive as their biggest challenges when trying to obtain nutritious foods on limited budgets. One solution is to provide direct subsidies to reduce the cost of healthy items or to form partnerships with farmers’ markets or supermarkets to make fresh produce more affordable. These measures could significantly enhance access to health-promoting foods in underserved communities [24].

In addition, future interventions should consider incorporating components that increase household income and promote financial literacy. Budgeting education, especially focused on meal planning and cost-effective shopping, may empower individuals to make healthier food choices despite economic strains. Programs that teach people how to compare prices, substitute ingredients wisely, and understand the long-term health value of certain foods can improve dietary behavior even when financial resources are limited.

### 7.3. Prostate Cancer and Nutritional Medical Recommendations: The Real World

When reviewing the international literature on cancer and nutrition, thousands of articles are available that explore positive and negative associations between specific foods and various types of cancer. However, despite the abundance of information, there is still no consensus regarding the direct role of nutrition in prostate cancer. In fact, the European Guidelines on Prostate Cancer report that there is no conclusive evidence to support a causal relationship between dietary patterns and the development of prostate cancer. As a result, no official nutritional recommendations or strategies have been established, even though numerous hypotheses have been proposed and widely debated [55].

A similar trend can be observed in the literature addressing lifestyle factors in prostate cancer prevention. Although several studies advocate for the adoption of healthy behaviors to reduce the risk of chronic diseases, their impact on clinical practice remains limited. Good lifestyle habits—such as balanced nutrition and physical activity—are encouraged for the general population, but not specifically emphasized in urology consultations or cancer prevention programs [4]. In clinical practice, lifestyle modifications are more frequently recommended after a prostate cancer diagnosis or during treatment, rather than as preventive strategies. Among these, regular resistance and aerobic exercise are commonly advised during active treatment, while nutritional guidelines receive comparatively little emphasis [53].

Clinicians today have access to a wealth of information regarding prostate cancer risk factors. Moreover, the accessibility of digital health information allows patients to become more informed and proactive about their disease, including adopting healthier lifestyles. This creates a favorable scenario for shared decision-making between patients and healthcare providers. Nevertheless, it remains uncertain how many clinicians systematically offer evidence-based lifestyle strategies to their patients. It would be of interest to investigate not only the frequency with which such recommendations are given but also their quality, scientific foundation, and degree of personalization.

### 7.4. Nutritional Education and Public Awareness

The public remains largely unaware of dietary guidance that can help prevent chronic diseases, revealing a widespread knowledge gap on an international scale. At-risk groups are often misinformed about the consumption of animal-derived foods and their association with prostate cancer risk. For consumers to adopt healthier practices, they need accurate information and a clear understanding of the benefits of specific dietary habits. Nutritional education strategies should focus on raising awareness of accessible, informed food options within communities. This approach integrates self-efficacy and empowerment as key factors influencing preventive dietary behaviors and emphasizes the role of social processes [3].

Effective strategies could include the development and dissemination of culturally relevant educational materials, delivered by nutrition educators in schools or community centers. Additionally, expert-led workshops at local supermarkets could address barriers to healthy eating by showcasing low-cost, easily accessible foods. Such initiatives could be marketed as a two-stage demonstration program in collaboration with community adult-education centers, ensuring a comprehensive and practical approach to promoting healthier eating habits [16].

In other international public health nutrition interventions, local mass media campaigns have increased community awareness, informed decision-making, and fostered discussion on physical activity. Community members reported feeling empowered by their new knowledge and expressed a need for supportive environments to sustain long-term change [16]. Coordinated efforts to provide consistent public health messages from multi-level agencies—including healthcare professionals—are critical for behavior change. Early childhood education in healthy lifestyles has also been advocated for preventing adult diet-related diseases, though its effectiveness specifically for prostate cancer prevention remains unproven. Finally, prompting the food industry to play a positive role in supporting healthier choices remains both a national and international public health priority. Where possible, the industry should be encouraged to contribute its own strategies, as it is an important stakeholder in dietary development [39].

## 8. Conclusions

Prostate cancer remains a significant public health challenge in Mexico, particularly among aging male populations with limited access to early screening and specialized care. This review highlights how modifiable lifestyle factors—especially diet—represent a promising avenue for reducing disease risk and improving outcomes. Scientific evidence supports the protective role of plant-based foods rich in antioxidants, fiber, and healthy fats—such as tomatoes, avocado, and whole grains—while also highlighting the adverse effects of processed meats, saturated fats, and refined sugars. Furthermore, culturally tailored dietary strategies grounded in Mexican culinary traditions can enhance adherence and facilitate the integration of preventive nutrition into daily life.

## Figures and Tables

**Figure 1 nutrients-17-02151-f001:**
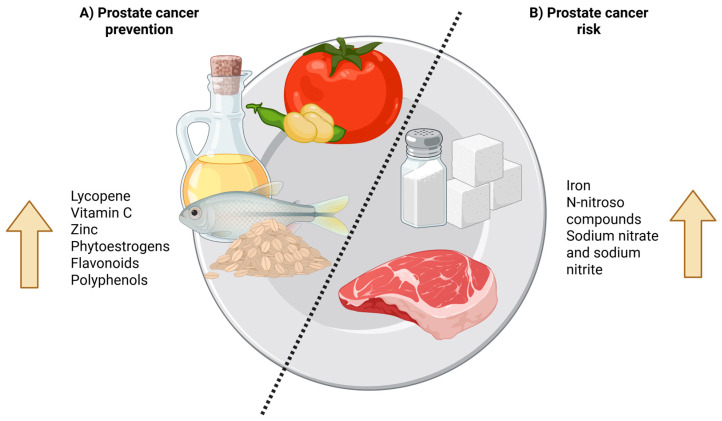
Nutritional effects on prostate cancer development and progression: (**A**) Protective foods thought to reduce the risk of progression of prostate cancer include tomato-based products, soy, vegetable oils, fish, and whole grains. These foods are rich in beneficial compounds such as lycopene, vitamin C, zinc, phytoestrogens, flavonoids, and polyphenols. (**B**) Harmful foods thought to increase the risk of prostate cancer include processed meats and refined sugars. These contain pro-carcinogenic substances such as iron, *N*-nitroso compounds, sodium nitrate, and sodium nitrite.

**Figure 2 nutrients-17-02151-f002:**
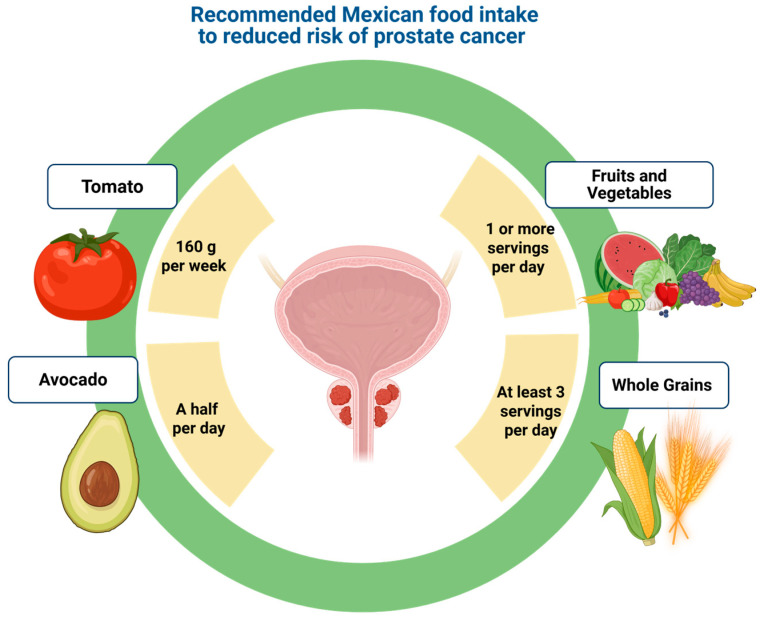
Recommended portions of traditional Mexican foods associated with a decreased risk of prostate cancer. The figure illustrates suggested servings of antioxidant-rich fruits and vegetables, fiber-dense whole grains, healthy fats such as avocado, and tomato-based products high in lycopene, all of which contribute to prostate cancer prevention within a culturally relevant dietary framework.

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
