# Peer review of "Nutrition and Diet in the Prevention and Management of Prostate Cancer in Mexico: A Narrative Review"

_nutrients, 2025, doi:10.3390/nu17132151_

Round 1
Reviewer 1 Report
Comments and Suggestions for Authors
Dear Authors,
Congratulations on an interesting idea for a topic for a paper. The data you collected allowed for an interesting way of presenting the research on the topic in question. It can be said that the described effect of diet on reducing the risk of prostate cancer - is not a new phenomenon, so the way in which you present this problem is important. It must be presented in an interesting way, especially since review papers are a “collection” of important information on a topic and the reader is supposed to benefit from the compiled knowledge, having read such an attribution. Obviously, the topic presented in this work, relating to the Mexican population, is a new element.
Therefore, I believe that the paper in this form has considerable shortcomings and needs to be rewritten. In my opinion, for such an important topic, the information and data it contains are treated too superficially. What I miss is a reference to other works by researchers dealing with the subject of the use of the discussed (lycopene, avocados, fruit and vegetables and others) potential components of the anti-cancer diet in terms of mechanisms of action or use in the diet. And there are a lot of such works (in 2025, I found 162 papers on lycopene or 32 on the use of avocados in preventive diets or cancer support therapy). Therefore, I think the work needs to be rewritten - although I will stress - the topic is extremely important and the work could be very interesting.
The chapters of the work are presented correctly and the content corresponds to the titles. Additional visualisation in the form of figures (graphical abstract) would have enriched this work.
I do not refer to grammatical or stylistic errors - as I am not competent in this area.
I hope that my comments will help you to prepare your amendments.
Author Response
Dear Authors,
Congratulations on an interesting idea for a topic for a paper. The data you collected allowed for an interesting way of presenting the research on the topic in question. It can be said that the described effect of diet on reducing the risk of prostate cancer - is not a new phenomenon, so the way in which you present this problem is important. It must be presented in an interesting way, especially since review papers are a “collection” of important information on a topic and the reader is supposed to benefit from the compiled knowledge, having read such an attribution. Obviously, the topic presented in this work, relating to the Mexican population, is a new element.
Therefore, I believe that the paper in this form has considerable shortcomings and needs to be rewritten. In my opinion, for such an important topic, the information and data it contains are treated too superficially. What I miss is a reference to other works by researchers dealing with the subject of the use of the discussed (lycopene, avocados, fruit and vegetables and others) potential components of the anti-cancer diet in terms of mechanisms of action or use in the diet. And there are a lot of such works (in 2025, I found 162 papers on lycopene or 32 on the use of avocados in preventive diets or cancer support therapy). Therefore, I think the work needs to be rewritten - although I will stress - the topic is extremely important and the work could be very interesting.
Answer: This is a key comment, we are thankful for this observation. Our research team agrees on the relevance of this topic in this review; therefore, a deeper search regarding the lycopene, avocados, fruit and vegetables, and others was performed and further information has been incorporated to prepare the final version. The sections 5.1 to 5.4 have been enriched and rewritten to cover more related studies:
- Line 441 to 479: We justify that lycopene has a strong antioxidant and anti-inflammatory potential in prostate cancer prevention.
- Line 545 to 562: For the avocado section we enhance the relevance in anti-inflammatory, pro-apoptotic, and antioxidants effects.
- Line 583 to 597: Antioxidants fruits and vegetables have potential compounds such as flavonoids and isoflavonoids that prevent prostate hyperplasia.
- Line 605 to 618: Grains and fiber it shown decrease inflammatory process.
The chapters of the work are presented correctly and the content corresponds to the titles. Additional visualization in the form of figures (graphical abstract) would have enriched this work.
Answer: Two figures have been incorporated in order to improve the presentation of the information: The first one focused on “Food effects in prostate cancer disease” line 288 and the second one on “Portions of Mexican foods associated with decreased risk of prostate cancer” line 641.
I do not refer to grammatical or stylistic errors - as I am not competent in this area.
Answer: We appreciate your comment. We would like to say that the English style has been double checked in order to polish the writing of the article.
I hope that my comments will help you to prepare your amendments.
Answer: Definitely your comments have been helpful to improve the quality of the paper, and we have taken them into account for modifications to be performed in the final version of the article.

Reviewer 2 Report
Comments and Suggestions for Authors
This manuscript reviews the role of nutrition and diet in the prevention and management of prostate cancer, with a specific focus on men living in Mexico. Both global and regional evidence of dietary risk are discussed, together with potential cultural barriers to dietary interventions. The manuscript highlights specific foods such as tomatoes, avocados, and whole grains, and their potential benefits in prostate cancer prevention.
This topic is relevant, particularly as prostate cancer is a leading cause of cancer-related deaths among men in Mexico. The authors present a well-referenced comprehensive review of epidemiological data, dietary risk and protective factors, and a useful inclusion of cultural and socioeconomic considerations with emphasis on practical dietary recommendations.
The manuscript cannot be recommended for publication in its present form as it requires major revision, specifically:
- It would be helpful to have clearer section and sub-sections, and to avoid duplication across sections.
- A Conclusion section is required (it currently contains text from the template).
- There are many grammatical and typographical errors. Here are a few examples:
- Line 33: "spread" not "spreads"
- Line 36: "populations" not "population"
- Line 43: "in some way which has been " can be deleted
- Lines 54 to 56: "Despite being one of the controversial topics today, it is not well known that in Mexico, prostate cancer is a significant and persistent problem in Mexico. " Needs to be rewritten.
- Line 68: "Although men are at greater risk of prostate cancer, " suggest "Although all men are at risk of prostate cancer, "
- There is a need to make a distinction between evidence obtained from observational and interventional studies.
- It would be helpful to have a table or figure explaining the main findings with some indication of the strength of evidence for each finding.
It would be extremely useful to explicitly identify areas where further research is needed, and describe the design of clinical trials (ideally randomised) that could address these questions. Ideally, these clinical trials should be "tailored" to the population of older men in Mexico.
In summary, the manuscript presents a valuable review of the role of diet in prostate cancer prevention and management in Mexico. With improvements described above, it has the potential to make a strong contribution to the field of nutritional oncology and public health.
Comments on the Quality of English LanguageThe quality of English language needs to be improved.
Author Response
This manuscript reviews the role of nutrition and diet in the prevention and management of prostate cancer, with a specific focus on men living in Mexico. Both global and regional evidence of dietary risk are discussed, together with potential cultural barriers to dietary interventions. The manuscript highlights specific foods such as tomatoes, avocados, and whole grains, and their potential benefits in prostate cancer prevention.
This topic is relevant, particularly as prostate cancer is a leading cause of cancer-related deaths among men in Mexico. The authors present a well-referenced comprehensive review of epidemiological data, dietary risk and protective factors, and a useful inclusion of cultural and socioeconomic considerations with emphasis on practical dietary recommendations.
The manuscript cannot be recommended for publication in its present form as it requires major revision, specifically:
- It would be helpful to have clearer section and sub-sections, and to avoid duplication across sections.
Answer: We appreciate the comment and we are open to modify the subsections if needed. We have carefully reviewed the titles and the order of the sections presented and we consider the information comprised in this review is presented in a logic order under the section titles reflecting the content of each one of them. However, we are open to restructure or modify it if needed, whether your consideration is still so. If that should be the case, we would like to have more information about this point in order to address this in a more proper manner.
- A Conclusion section is required (it currently contains text from the template).
Answer: Thank you for suggesting to add a conclusion of the narrative review; we consider something really important to close the work with a short highlight of the main ideas and the perspectives to this topic. (Lines 962-982)
- There are many grammatical and typographical errors. Here are a few examples:
- Line 33: "spread" not "spreads"
- Line 36: "populations" not "population"
- Line 43: "in some way which has been " can be deleted
- Lines 54 to 56: "Despite being one of the controversial topics today, it is not well known that in Mexico, prostate cancer is a significant and persistent problem in Mexico. " Needs to be rewritten.
- Line 68: "Although men are at greater risk of prostate cancer, " suggest "Although all men are at risk of prostate cancer, "
Answer: We really appreciate this observation. The forementioned examples were modified and a deeper review of the English style was performed to improve the writing. If needed, we are open to submit the article to a professional style revision.
- There is a need to make a distinction between evidence obtained from observational and interventional studies.
Answer: We appreciate the reviewer’s suggestion and acknowledge the importance of distinguishing between observational and interventional evidence in systematic or meta-analytic approaches. However, given that this is a narrative review aimed at providing a broad overview of the scientific literature on nutrition and prostate cancer in the Mexican context, we chose to integrate findings from both study types to emphasize general patterns, mechanisms, and public health implications. We believe this integrative approach aligns with the aims of the review and serves the intended audience without compromising scientific rigor. Nonetheless, we have clarified the nature of key studies when particularly relevant to the strength of the evidence.
- It would be helpful to have a table or figure explaining the main findings with some indication of the strength of evidence for each finding.
Answer: Two figures have been incorporated in order to improve the presentation of the information: The first one focused on “Food effects in prostate cancer disease” line 288 and the second one on “Portions of Mexican foods associated with decreased risk of prostate cancer” line 641.
It would be extremely useful to explicitly identify areas where further research is needed, and describe the design of clinical trials (ideally randomised) that could address these questions. Ideally, these clinical trials should be "tailored" to the population of older men in Mexico.
Answer: We appreciate this very important comment. We have incorporated this information in the conclusion which highlights the multidisciplinary research needed to can support early prevention and management of prostate cancer. This interdisciplinary collaboration across healthcare, government, and community sectors is essential to address both biological and social determinants of health.
In summary, the manuscript presents a valuable review of the role of diet in prostate cancer prevention and management in Mexico. With improvements described above, it has the potential to make a strong contribution to the field of nutritional oncology and public health.
Reviewer 3 Report
Comments and Suggestions for Authors
In the present paper, Sarai C. Rodríguez Reyes and colleagues analyzed global scientific evidence on the role of diet in preventing and managing prostate cancer, while also considering the social, economic, and cultural factors affecting its implementation in the Mexican population. The authors suggest that, in the light of the data obtained, it is necessary to empower and create awareness about prostate cancer in individuals and the general population, then promoting an appropriate diet that decreases their risk of prostate cancer. Overall, I think that the manuscript is informative (within the scope of "Nutrients”) and of clinical impact on a current topic of interest. So far, in my humble opinion, I would make a series of specific points to address carefully in order to improve the overall quality of manuscipt.
1) Please underline in the title and in appropriate sections of revised paper that the present review is a “narrative review”.
2) There is growing evidence that the microbiome is involved in the development and treatment of many human diseases, including prostate cancer. Please deeply discuss this very intriguing topic of current research (for your convenience see: Prostate Cancer Prostatic Dis 2022, 25, 159-164; Int J Mol Sci. 2023, 24, 2, 1511).
3) The Serenoa repens, lycopene and selenium nutraceutical association may have greater potential for the management of benign prostate hyperplasia, also potentially reducing prostate cancer risk (see: Curr Med Chem. 2013, 20, 10, 1306-12; Arch Ital Urol Androl. 2019, 2, 91, 3; Nutrients. 2020, 12, 10, 2985). Please deeply discuss this trending topic for current research.
4) The authors could add in Graphical form (informative tables, figures etc.) the different points discussed in the present manuscript (for example, specific foods and their potential benefits, healthy dietary habits suggested, etc.). As a matter of fact, I feel that the readers can better understand the potential preventive/therapeutic role of the healthy diet in preventing and managing prostate cancer.
5) In the revised version of paper, it would be appropriate to shorten the discussion section, also adding a conclusion paragraph that better emphasizes the main objectives of the present narrative review.
Author Response
In the present paper, Sarai C. Rodríguez Reyes and colleagues analyzed global scientific evidence on the role of diet in preventing and managing prostate cancer, while also considering the social, economic, and cultural factors affecting its implementation in the Mexican population. The authors suggest that, in the light of the data obtained, it is necessary to empower and create awareness about prostate cancer in individuals and the general population, then promoting an appropriate diet that decreases their risk of prostate cancer. Overall, I think that the manuscript is informative (within the scope of "Nutrients”) and of clinical impact on a current topic of interest. So far, in my humble opinion, I would make a series of specific points to address carefully in order to improve the overall quality of manuscipt.
- Please underline in the title and in appropriate sections of revised paper that the present review is a “narrative review”.
Answer: Narrative Review was incorporated in the title (Lines 2-3)
- There is growing evidence that the microbiome is involved in the development and treatment of many human diseases, including prostate cancer. Please deeply discuss this very intriguing topic of current research (for your convenience see: Prostate Cancer Prostatic Dis 2022, 25, 159-164; Int J Mol Sci. 2023, 24, 2, 1511).
Answer: We appreciate this comment, which is a key suggestion because of the relevance of the microbiome in the prostate cancer. A deep research has been conducted, and a complete section was incorporated (Section 6) named: “The Gut Microbiome-Prostate Cancer Axis: Mechanisms and Nutritional Modulation” (Lines 644-794)
- The Serenoa repens, lycopene and selenium nutraceutical association may have greater potential for the management of benign prostate hyperplasia, also potentially reducing prostate cancer risk (see: Curr Med Chem. 2013, 20, 10, 1306-12; Arch Ital Urol Androl. 2019, 2, 91, 3; Nutrients. 2020, 12, 10, 2985). Please deeply discuss this trending topic for current research.
Answer: A deep research was performed over these articles and 6 paragraphs have been added in the lycopene section discussing the lycopene and selenium association to PCa risk (Lines 480-528).
- The authors could add in Graphical form (informative tables, figures etc.) the different points discussed in the present manuscript (for example, specific foods and their potential benefits, healthy dietary habits suggested, etc.). As a matter of fact, I feel that the readers can better understand the potential preventive/therapeutic role of the healthy diet in preventing and managing prostate cancer.
Answer: Two figures have been incorporated in order to improve the presentation of the information: The first one focused on “Food effects in prostate cancer disease” line 288 and the second one on “Portions of Mexican foods associated with decreased risk of prostate cancer” line 641.
- In the revised version of paper, it would be appropriate to shorten the discussion section, also adding a conclusion paragraph that better emphasizes the main objectives of the present narrative review
Answer: Thank you for suggesting to add a conclusion of the narrative review; we consider something really important to close the work with a short highlight of the main ideas and the perspectives to this topic. As for the extension of the discussion, we are thankful for the suggestion, however, we consider it is important to deeply cover the studied topics in order to have a broad view of the entire landscape around the diverse sections. (Lines 962-982)
Round 2
Reviewer 1 Report
Comments and Suggestions for Authors
Dear Authors,
Once again, congratulations on an interesting idea for a topic for a paper. Thank you very much for taking the time to address my comments and indeed for heavily redrafting your paper. I also believe that the enrichment of the data by adding important information contained in other works not cited by you (an increase of 14 literature items) and the introduction of Figures 1 and 2. has definitely improved the quality of the paper. I therefore believe that the work can be published in this form. I wish you continued success.
Comments on the Quality of English LanguageI do not refer to grammatical or stylistic errors - as I am not competent in this area.
Reviewer 2 Report
Comments and Suggestions for Authors
The authors have made some but not all required revsions to the manuscript, specifically:
- There is still duplication across section, which is probably best dealth with by having clearer sections and subsections. For example, "tomatoes" are described several places in sections 3 and 5.
- The first paragraph of the Conclusions section is good, but the second paragraph should be deleted as it imples the current evidence is strong.
- The English has been improved somewhat, but the manuscript would benefit from a professional revision.
- Interventional versus Observational evidence. The new sections specifically addresses interventional studies, which is great. However, the use of the word "intervention" elsewhere is misleading (e.g. lines 125, 254, 676, 679, 792). Much of the evidence for changing the diet to reduce risk comes from observational studies, and this should be made very clear.
- The figures are helpful, but Figure 1 is a bit confusing. Perhaps label as "thought to reduce risk" and "thought to increase risk". Also, shouldn't all of the components of Figure 2 be shown on the plate?
Comments on the Quality of English Language
The quality of English language needs to be improved.
Author Response
The authors have made some but not all required revsions to the manuscript, specifically:
- There is still duplication across section, which is probably best dealth with by having clearer sections and subsections. For example, "tomatoes" are described several places in sections 3 and 5.
Answer: We appreciate the reviewer’s thoughtful observation. We acknowledge that certain foods, such as tomatoes, are mentioned in more than one section. This was intentional, as the earlier sections (e.g., 3.2) provide a general overview of their nutritional role and scientific relevance, while Section 5 focuses on culturally tailored, food-specific recommendations for the Mexican population. However, to improve clarity and avoid redundancy, we have revised the structure and wording in both sections to reduce overlap and enhance the distinction between general evidence and region-specific applications. We hope these changes address the concern and improve the manuscript’s readability.
- The first paragraph of the Conclusions section is good, but the second paragraph should be deleted as it imples the current evidence is strong.
Answer: As suggested, the second paragraph was deleted from the conclusion leaving only the first paragraph.
- The English has been improved somewhat, but the manuscript would benefit from a professional revision.
Answer: A professional English revision was performed and the certificate is attached. We consider the quality of the style in the writing for this new version is higher and we would be grateful if this is also perceived by the reviewer; however in the case this new version´s English style does not meet the quality expected we would be thankful to know what sections in particular can be improved.
- Interventional versus Observational evidence. The new sections specifically addresses interventional studies, which is great. However, the use of the word "intervention" elsewhere is misleading (e.g. lines 125, 254, 676, 679, 792). Much of the evidence for changing the diet to reduce risk comes from observational studies, and this should be made very clear.
Answer: We thank the reviewer for this valuable observation. In response, we carefully reviewed the use of the term “intervention” throughout the manuscript. We retained the term only in contexts referring to actions implemented (or potentially implementable) in patient populations. We acknowledge the concern regarding its previous use in relation to observational evidence and have revised the text accordingly to avoid misinterpretation. Should any instances still appear ambiguous, we would appreciate further guidance and will be glad to make additional adjustments as needed.
- The figures are helpful, but Figure 1 is a bit confusing. Perhaps label as "thought to reduce risk" and "thought to increase risk". Also, shouldn't all of the components of Figure 2 be shown on the plate?
Answer:
We sincerely thank the reviewer for this valuable observation and for their careful assessment of our work.
In the description of the figure 1 the phrases "thought to reduce risk" and "thought to increase risk" were added to make the explanation clearer. As for figure 2 we consider that not necessarily the components should be shown on the plate, however, the description of the figure was rewritten to make it more descriptive of the content of the image. We hope the images and explanation for each one are clearer as they are now presented.
The foods represented in Figure 2 were selected based on two main criteria: (1) the available scientific evidence supporting their potential protective effect against prostate cancer, and (2) their cultural and dietary relevance in the context of the Mexican population. Specifically, we prioritized foods that are staples of the traditional Mexican diet, widely accessible across different regions, and economically affordable for the general population. These include tomatoes, avocados, a variety of fruits and vegetables, and whole grains such as corn, which are not only commonly consumed but also rich in bioactive compounds with potential chemopreventive properties.
While we acknowledge the importance of illustrating a more comprehensive dietary pattern, the figure was intentionally simplified to enhance clarity and to reflect realistic and culturally appropriate dietary recommendations. Including a broader array of foods might have compromised the visual impact and readability of the image for public health communication, which is one of the objectives of this figure.
Nevertheless, we appreciate the suggestion and have noted it as a potential enhancement for future visual materials or supplementary content, should the journal require a more detailed version.
Thank you again for your insightful feedback

Reviewer 3 Report
Comments and Suggestions for Authors
Thank you for addressing my comments well. I have no further remarks.
Author Response
Thank you for addressing my comments well. I have no further remarks.
Thank you for your approval
